# Characterization of *Bacillus pumilus* Strains Isolated from Bovine Uteri

**DOI:** 10.3390/ani13081297

**Published:** 2023-04-10

**Authors:** Panagiotis Ballas, Christoph Gabler, Karen Wagener, Marc Drillich, Monika Ehling-Schulz

**Affiliations:** 1Functional Microbiology Unit, Institute for Microbiology, Department of Pathobiology, University of Veterinary Medicine, 1210 Vienna, Austria; 2Clinical Unit for Herd Health Management in Ruminants, Department for Farm Animals and Veterinary Public Health, University of Veterinary Medicine, 1210 Vienna, Austria; 3Department of Veterinary Medicine, Freie Universität Berlin, Institute of Veterinary Biochemistry, 14163 Berlin, Germany

**Keywords:** endometritis, *Bacillus pumilus*, pathogenicity, cell infection assays

## Abstract

**Simple Summary:**

Bound with proper productivity, reproductive performance in dairy cows is of utmost importance. One of the most important conditions hampering the reproductive performance of a dairy cow is bovine endometritis. In this paper, we aim to report the effect of *B. pumilus* strains, isolated from cows with or without endometritis, with regard to their pathogenicity to the bovine uterus by using a cell-culture-based approach. Our study revealed that *B. pumilus* strains possess specific genes related to enzyme production and harm the endometrial bovine cells in a dose-dependent pattern.

**Abstract:**

Uterine infections are a major source of economic losses to dairy farmers. The uterine microbiota as well as opportunistic uterine contaminants can contribute to the development of endometritis in dairy cows during the postpartum period. Therefore, it is important to characterize potential pathogens and to further elucidate their role in the disease. In this study, we aimed to characterize *Bacillus pumilus* field isolates to obtain more details regarding their effect on uterine cells by using an in vitro endometrial epithelial primary cells model. We found that *B. pumilus* isolates possessed the keratinase genes *ker1* and *ker2* and therefore may produce keratinases. When primary endometrial epithelial cells were infected with 4 different *B. pumilus* strains, an effect on cellular viability was observed over the course of 72 h. The effect was dose-dependent and time-dependent. Nevertheless, significant differences between the strains were not observed. All tested strains reduced the viability of the primary cells after 72 h of incubation, indicating that *B. pumilus* potentially has a pathogenic effect on endometrial epithelial cells.

## 1. Introduction

Recent opinions pinpoint that the development of postpartum uterine diseases in cows may have three major components: avoidance, tolerance and resistance [1]. Therefore, the development of uterine diseases, such as endometritis, is affected by multiple factors. One of the most important of those factors is the uterine microbiota and the postpartum contamination of the uterus with bacteria [2]. Generally, the bovine uterus was considered sterile outside of the postpartum period, but newer studies debate that opinion [3,4]. Over the last decade more information regarding pathogenic bacteria and the postpartum uterine microflora in dairy cows has surfaced, as the use of both culture-dependent and culture-independent techniques provided a wealth of information regarding the postpartum uterine microbial community [5,6,7,8,9,10,11,12]. Under the current opinions, the bacteria involved in bovine endometritis can be categorized into three groups: defined pathogens, potential pathogens and opportunistic colonizers of the uterus [13]. Defined uterine pathogens, such as *Trueperella pyogenes* and *Escherichia coli*, have been thoroughly studied in the past, providing evidence regarding their role in the disease [14,15,16,17]. On the contrary, bacteria such as *Streptococcus uberis*, which are considered as potential pathogens, were only rarely studied regarding their role in endometritis [8,18], and only recently have new studies elucidated their potentially harmful effect on uterine cells [19]. Bacterial species considered as opportunistic contaminants of the uterus, such as *Bacillus pumilus*, were just recently brought to the attention of researchers regarding their potential role in the development of endometritis and their effect on uterine cells [20], even though many of them possess pathogenic molecular mechanisms, such as the production of potentially cell-harmful enzymes or toxins [21]. Nevertheless, the abovementioned categorization is not absolute, and studies which will clearly define and characterize the pathogenicity of field-isolated bacterial isolates are needed.

This scope of this study is to characterize case-isolated *B. pumilus* strains isolated from bovine uteri during the postpartum period, with a focus on strains isolated from animals with endometritis, and to evaluate the potential effect of these strains on the viability of endometrial cells.

## 2. Materials and Methods

### 2.1. Bacterial Strains Culture and Growth Kinetics

The strains were collected in a previous study [18]. A total of 14 strains from 40 Holstein-Friesian cows sampled and monitored throughout the first days of the postpartum period were selected. The samples were collected from animals showing clear symptoms of endometritis (vaginal discharge score 3; 19/40 cows) and animals without endometritis, by using the cytobrush technique [22]. Bacteria were revived from glycerol stocks at −80 °C and were recovered by streaking onto Columbia III agar supplemented with 5% sheep blood (Becton Dickinson, Heidelberg, Germany). Bacteria were incubated at 37 °C for 24 h under aerobic conditions. Bacteria were grown routinely for the analyses in Columbia III agar supplemented with 5% sheep blood at 37 °C for 24 h.

For co-culturing, endometrial epithelial cells and bacteria were prepared as before [20] with slight modifications. Bacteria were primarily cultured in brain heart infusion (BHI) broth (Sigma-Aldrich, Taufkirchen, Germany) at 37 °C for 24 h, under aerobic conditions. The bacterial suspension was centrifuged at 2000× *g* for 10 min, and the bacteria were collected from the solution. The bacterial pellet was washed once with Dulbecco’s phosphate-buffered saline (PBS, Biochrom, Berlin, Germany), re-suspended in BHI broth with 20% glycerol *v*/*v* and stored at −80 °C. Plate counting on BHI agar (Sigma-Aldrich) was performed to determine the number of colony-forming units (CFU) per ml. Bacteria were thawed briefly before the start of the experiments and serially diluted in epithelial cell culture medium without antibiotics (DMEM/Ham’s F-12 +10% fetal bovine serum (FBS), all from Biochrom) to obtain the required bacterial concentrations for each assay. For the growth kinetics, bacteria were cultivated from a starting inoculum of 103 bacterial cells in DMEM/Ham’s F-12, supplemented with 10% FBS for 48 h at 37 °C, 5% CO_2_. Optical density was measured every 4 h for the first 24 h, and every 8 h from 24 to 48 h. Optical density (OD600) was measured in a Jenway 6305 spectrophotometer (Stone, Staffordshire, UK).

### 2.2. Keratinase Gene PCR

Evaluation of the presence of the keratinase genes *ker1* and *ker2* was performed as described before by [21], with slight modifications. For the PCR reaction, a mixture consisting of 1 μM of the respective oligonucleotide primers, 0.4 μΜ dNTPs, 2.5 mM MgCl2, 1 × GoTaq Reaction Buffer and 0.75 U goTaq polymerase (Promega, Mannheim, Germany) was used, along with 1 μL template DNA.

The PCR protocol started with an initial denaturation step for 2 min at 95 °C, followed by 30 cycles of: denaturation at 95 °C for 35 s, annealing for 30 s at appropriate temperature and extension at 72 °C, with the time depending on the length of the amplicon. A final extension step of 4 min at 72 °C was performed before the products were cooled at 4 °C. The presence of the specific bands was checked by gel electrophoresis.

### 2.3. Primary Endometrial Epithelial Cell Culture

Primary endometrial epithelial cell culturing was carried out as described before [18,23]. Briefly, bovine uteri were collected from a local slaughterhouse, and small tissue pieces (approximately 0.1 cm^2^) from the area between the caruncles were collected. The collected tissue was minced with scalpels and then digested for 2 h at 37 °C in a solution containing 150 U/mL of collagenase, 150 U/mL of hyaluronidase, 200 U/mL of penicillin and 20 μg/mL of streptomycin (Sigma-Aldrich) in Hank’s balanced salt solution (Biochrom). Cells were then centrifuged, and the cell pellet was washed with cell culture medium (DMEM/Ham’s F-12 and 10% FBS, gentamicin and amphotericin B; all from Biochrom). The cells were seeded in 25 cm^2^ flasks at 37 °C + 5% CO_2_ for 18 h. During this time, fibroblast cells had already attached, while the medium containing non-attached cells was transferred to a new 25 cm^2^ flask in order to obtain a pure (>99%) epithelial cell culture. Cells were transferred to a 75 cm^2^ flask when a confluence level of >80% was achieved. Cells were allowed to grow and finally passaged into a 24-well plate at a final density of 2 × 10^5^ cells per well.

### 2.4. Viability Assay

Primary endometrial endothelial cells were infected with different multiplicity of infection (MOI) concentrations of *B. pumilus* and then incubated at 37 °C, 5% CO_2_ for 72 h. Evaluation of the cellular viability was carried out by using 0.5% trypan blue (ScienCell, CA, USA) staining as described before [20]. Cellular viability was determined at 24, 48 and 72 h post-inoculation. For calculating the viability, multiple optical fields were evaluated, and the numbers of unstained cells (viable) and stained cells (non-viable) were calculated as percentages of the total number of cells.

### 2.5. Statistical Analysis

For the analysis of data and for data visualization, GraphPad Prism version 7.00 for Windows, by GraphPad Software (La Jolla, San Diego, CA, USA), was used. For the growth curve data, a Shapiro–Wilk test was used in order to test the normality of the data as it is more fitted to smaller sample sizes. Comparisons were performed by using the Welch’s test. For the cell viability assay data, a Wilcoxon signed rank test was performed, as we were comparing cellular viability scores. A *p*-value of <0.05 was considered as significant, while *p*-values from 0.05 to 0.1 were considered as a trend.

## 3. Results

### 3.1. Keratinase Gene Screening, and Growth Kinetics

The keratinase genes screening was performed as described above. In the pool of tested strains, all strains were found positive for the *ker1* gene, while only 1 out of 14 strains was positive for the *ker2* gene. Because no specific pattern of gene presence between strains originating from healthy and diseased animals was found, *B. pumilus* strains originating from animals diagnosed with endometritis at day 21 postpartum were selected for further study. First, the growth of the bacteria in DMEM/Ham’s F-12 supplemented with 10% FBS was monitored. The four strains BP6, BP10, BP13 and BP18 showed generally similar growth patterns in the cell culture medium, as depicted in Figure 1.

Nevertheless, strain BP10 exhibited higher growth after 24 h compared to the strains BP6, BP13 and BP18, and after 32 h there was a significantly higher growth of strain BP10 compared to the strains BP13 and BP18 (*p* = 0.002 and *p* = 0.018, respectively), but also a delayed onset of the exponential growth phase.

### 3.2. Effect of B. pumilus Isolates on Uterine Cells

*B. pumilus* strains prepared as indicated before were used to infect the primary endometrial epithelial cells. The viability of the cells was evaluated at 24, 48 and 72 h after inoculation. After 24 h of incubation, minimal impairment of cellular viability was observed, with a cellular viability of approximately 95% in MOI 1 (Figure 2).

Higher MOI had a stronger effect on reducing cellular viability. Cellular deterioration gradually increased over time, with cellular viability being reduced to approximately 80% for strain BP6, 60% for the strains BP10 and BP13 and 40% for BP18 in MOI 1 after 48 h (Figure 2). Cellular viability decreased further at 72 h post-inoculation, with the exception of cells infected with the strain BP10, in which the viability did not reduce further. In addition, no strain-specific differences in the killing capacity of the isolates were observed. Additionally, the color of the medium was monitored throughout the 72 h period. No changes were observed, indicating that there were no secondary metabolites or medium changes (such as pH drop). Therefore, it is assumed that the observed cytotoxic effect can be attributed to the presence of the bacterium itself, rather than to other factors related to medium changes or medium nutrient depletion. Furthermore, during the time window it was observed that the tested *B. pumilus* strains caused detachment of the uterine cells, as depicted in Figure 3.

## 4. Discussion

Information regarding the role of the uterine microbiota and invasive bacterial species in bovine endometritis has largely expanded over the last several years. In the past, the uterus was considered as a sterile environment before parturition, but recent studies have indicated that it harbors its own unique microbiome with a broad variety of bacterial species [3]. The microbiome and opportunistic pathogens of the uterus have a dynamic relationship which may contribute to the development of the disease. The bacteria which can be related to endometritis in dairy cows can be categorized as pathogens, potential pathogens and opportunistic contaminants [9]. Nevertheless, it is important to characterize the role of each bacterial species with robust in vivo and in vitro models. To this end, we sought to characterize *B. pumilus* field isolates collected during previous studies [18]. The selected field isolates were subjected to screening for the keratinase genes *Ker1* and *Ker2*. Until now, no other study has considered a screening of *B. pumilus* isolates from a farm environment for the presence of the aforementioned genes, even though Bacillus spp. have the potential for production of keratinases. All strains tested positive for the *Ker1* gene, but only one was positive for *Ker2*. Therefore, it is assumed that the strains possess the ability to produce at least one type of keratinase. *B. pumilus* keratinases are considered to be subtilisin-like serine proteases [21] and have been considered as a potential virulence factor in various fungi species, especially dermatophytes such as *Microsporum canis* [24]. Furthermore, recent studies in mice models have indicated that certain keratin types are present in the endometrium and could potentially be related to reproductive performance and decreased conception rates [25]. Therefore, it is tempting to speculate that the presence of bacterial keratinases could have an effect on the tissue keratin, thus reducing tissue resistance.

Additionally, previous studies have indicated that *B. pumilus* can increase the mRNA expression of certain pro-inflammatory factors in primary endometrial cells [20]. In our study, an effect on cellular viability was observed for all the *B. pumilus* strains tested. Although specific differences between strains were not found, it was observed that all the tested strains had the ability to reduce the viability of endometrial primary cells. This effect is in line with previous studies indicating that *B. pumilus* has a destructive effect on endometrial cells [20]. By measuring the growth of the strains in the cell culture medium under the same conditions used for the cell viability assay, it was ensured that the effect of *B. pumilus* on cellular viability could be attributed to the presence of the bacterium and not to secondary changes of the medium, such as pH change or the production of bacterial metabolites. Nevertheless, more studies will be needed to observe more precisely what happens at the bacterial-cellular level regarding the metabolites and the cytotoxic effect.

## 5. Conclusions

In summary, our study provides initial results on *B. pumilus* field isolates’ *cytotoxicity* towards endometrial cells, indicating a potential contribution of *B. pumilus* to bovine endometritis. Certain isolates were able to strongly reduce the cellular viability of uterine cells when tested in a cell-culture model, but extreme differences in the pathogenicity were not observed. The pathogenic effect shown is possibly attributed to the bacterial cell itself rather than to secondary metabolic products of the bacteria; however, further studies will be necessary to decipher the factors involved in the observed cytotoxic effects of *B. pumilus* on endometrial cells.

## Figures and Tables

**Figure 1 animals-13-01297-f001:**
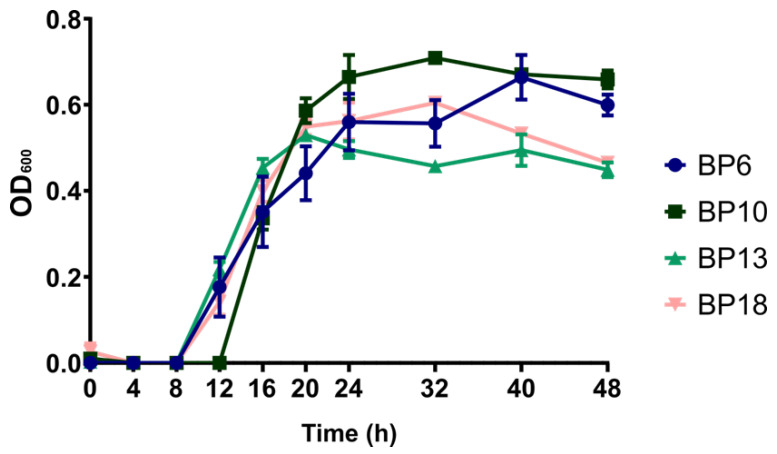
Growth profiles of the four selected *B. pumilus* strains. OD_600_ = optical density measured at 600 nm. Bars indicate standard error of mean.

**Figure 2 animals-13-01297-f002:**
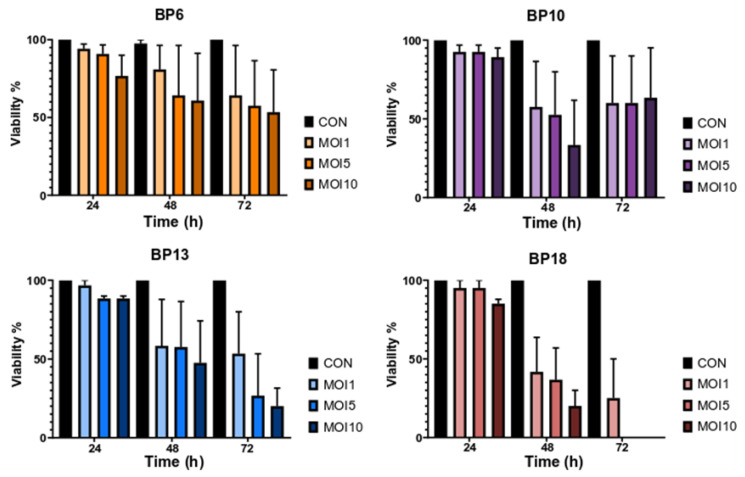
Viability of primary endometrial epithelial cells after being infected with *B. pumilus* endometritis isolates. Bars represent viability, and error bars represent the standard error of mean.

**Figure 3 animals-13-01297-f003:**
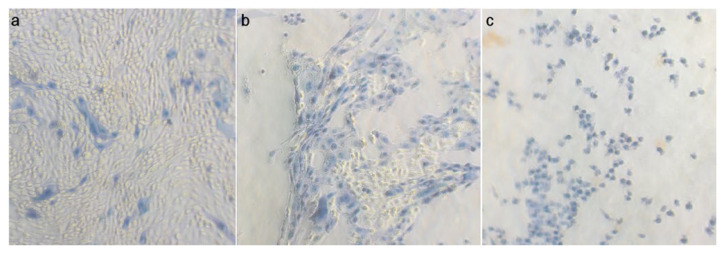
Staining of endometrial epithelial cells with trypan blue after co-culturing with *B. pumilus* strain BP13 at MOI 5: (**a**) at 24 h; (**b**) at 48 h; (**c**) at 72 h post-infection. Cellular detachment is evident after 48 h.

## Data Availability

Not applicable.

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
