# Peer review of "Characterization of Bacillus pumilus Strains Isolated from Bovine Uteri"

_animals, 2023, doi:10.3390/ani13081297_

Round 1

Reviewer 1 Report

The author isolated B. pumilus from cow with or without endometritis, and detected the effect the pathogenicity to the bovine uterus of B. pumilus by using a cell-culture-based approach. The idea is worth studying, but the content is not enough

1.      Can B. pumilus only be detected in the uterus of cows with endometritis? What percentage is detected? How many sample sizes were counted?

2.      What is the reason for choosing B. pumilus based on ker1 and ker2 genes? Are ker1 and ker2 specific to the B. pumilus or do other bacteria have the gene?

3.      The effect of B. pumilus on endometrial cell viability alone does not indicate its pathogenicity.

Author Response

Dear reviewer,

We would like to thank you reviewing our manuscript and your constructive comments which helped us to improve our scientific output. Please find our responses below.

Reviewer Nr 1

The author isolated B. pumilus from cow with or without endometritis, and detected the effect the pathogenicity to the bovine uterus of B. pumilus by using a cell-culture-based approach. The idea is worth studying, but the content is not enough

>1.      Can B. pumilus only be detected in the uterus of cows with endometritis? What percentage is detected? How many sample sizes were counted?

  1. pumilus can be retrieved from the uterus of both cows with or without endometritis. In order to address your comment we corrected paragraph 2.1 in Materials and Methods and explained how many animals were enrolled at the initial sampling, and what percentage of animals showed signs of endometritis.

>2.      What is the reason for choosing B. pumilus based on ker1 and ker2 genes? Are ker1 and ker2 specific to the B. pumilus or do other bacteria have the gene?

The ker1 and ker2 genes are specific for Bacillus pumilus. We opted for choosing to test these genes as it might be that they have a potential role in the pathogenicity. Recent studies (Reference 25: Zhang et al, 2020) have shown that specific keratin molecules (Keratin 86) can be determinants of uterine receptivity and success of embryo implantation. Also hormonal control regulates the expression of keratin in uterine cells. Therefore it was tempting to check whether there is any specific pattern with keratinase producing Bacillus pumilus regarding cellular viability and effect on the cells.

Please have in mind that the manuscript is a Short communication with limited length, therefore we cannot go to much into details.

>3.      The effect of B. pumilus on endometrial cell viability alone does not indicate its pathogenicity.

We agree to that point with the reviewer. Further experiments to elucidate the mechanisms are needed before making statements regarding pathogenicity (as the term may imply as well a contribution to the development of disease). Therefore we changed the text throughout the paper to “effect on uterine cells” or “effect on cellular viability”. 

Reviewer 2 Report

This paper approaches a relevant issue. However, the English needs a  review. In addition, there is a need to improve fluency of the text, especially regarding the connection between adjacent ideas. Overall, readability is low and requires attention from the authors

·      Please state a clear objective for the work in the abstract as well as in the main text. Try to improve the clarity of the introduction. 

·      Please add new references as: 

doi: 10.3389/fphar.2016.00382. 

Author Response

Dear reviewer,

We would like to thank you reviewing our manuscript and your constructive comments which helped us to improve our scientific output. Please find our responses below.

Reviewer Nr 2

Comments and Suggestions for Authors

>This paper approaches a relevant issue. However, the English needs a review. In addition, there is a need to improve fluency of the text, especially regarding the connection between adjacent ideas. Overall, readability is low and requires attention from the authors

We have done our best to improve the connection between ideas, and concepts of the paper. The manuscript was sent to a proof reading service and additionally we proof read the paper once more in order to find grammatical or spelling mistakes.

> Please state a clear objective for the work in the abstract as well as in the main text. Try to improve the clarity of the introduction.

A clear objective was added to the abstract and introduction, the text was revised accordingly.

> Please add new references as:

doi: 10.3389/fphar.2016.00382.

The suggested reference is an interesting paper; however, we can not see the relation to our manuscript and where to add it in the text. Therefore we opted not to use the suggested reference.

Reviewer 3 Report

In general, the work is good. These are preliminary results of a more comprehensive study.

Some changes are necessary, lack of rigor in terms of scientific writing, which must be corrected.

Attached is the file with the changes and suggestions.

The content of the attached file:

Abstract

L28 – 29 - The following sentence “The effect was dose-dependent and time-dependent, although significant differences between the strains were not observed” needs to be rewritten, when it says "although significant differences" it is necessary to put two parameters.

Introduction

L58 – Uteri?? I think they mean “uterus”

Observation: In the introduction, as a rule, a synthesis should be made of what led us to do the work and see what exists in the most recent bibliography. Of the references used in the introduction, the most recent one is from 2020, the others are all previous, most of which are more than 5 years old.

They should revise the introduction, including more current references.

Material and methos

L109 – “final density of 2 x 105 cells per well” modify to “2 x 105 cells per well”

L112 – “CO2” modify to “CO2”. Must modify throughout entire manuscript.

You should write in more detail why you use these statistical tests. Has the homogeneity of the data been checked?

Results

Figure 1 -  The figure caption should only contain information about the graph, not referring to  anything concrete. These remarks should only be placed in the text of the results and/or discussion. They should indicate what the bars shown in fig. Are they standard deviation bars or SEM?

Discussion

The last paragraph of the discussion is a conclusion and is practically the same as the conclusion of the work. It should be removed from the discussion.

Conclusion

The conclusion is good, but it must respond clearly to the proposed objective.

Author Response

Dear reviewer,

We would like to thank you reviewing our manuscript and your constructive comments which helped us to improve our scientific output. Please find our responses below.

Reviewer Nr 3

>Abstract

L28 – 29 - The following sentence “The effect was dose-dependent and time-dependent, although significant differences between the strains were not observed” needs to be rewritten, when it says "although significant differences" it is necessary to put two parameters.

We have corrected the following sentence and split it into two sentences in order to make it more easily readable.

>Introduction

L58 – Uteri?? I think they mean “uterus”

Uteri is the plural form of uterus. In this context the plural form is needed as the strains have been collected from the uterus of various cows, therefore in this context “uteri”.

>Observation: In the introduction, as a rule, a synthesis should be made of what led us to do the work and see what exists in the most recent bibliography. Of the references used in the introduction, the most recent one is from 2020, the others are all previous, most of which are more than 5 years old.

The introduction has been revised, and re-written and newer references were added. Nevertheless, we should state that as this area of research is relatively new, there are not a lot of studies published yearly and we opt to use studies that have been well established from groups that are well renowned on the field.

>Material and methods

>L109 – “final density of 2 x 105 cells per well” modify to “2 x 105 cells per well”

The sentence was corrected.

>L112 – “CO2” modify to “CO2”. Must modify throughout entire manuscript.

Corrected throughout the manuscript

>You should write in more detail why you use these statistical tests. Has the homogeneity of the data been checked?

We have revised this part and explained tests shortly. Please have in mind that the manuscript is a Short communication with limited length. We tested our growth curve data for normality with a Shapiro-Wilk test. The homogeneity of the data was not checked, as we opted to test for the data normality which is to our opinion more valuable when it comes to data related to growth measurements (optical density)

>Results

Figure 1 -  The figure caption should only contain information about the graph, not referring to  anything concrete. These remarks should only be placed in the text of the results and/or discussion. They should indicate what the bars shown in fig. Are they standard deviation bars or SEM?

The legend of the figure was modified in order to have the required information. The information regarding the bars was added to the legend.

>Discussion

The last paragraph of the discussion is a conclusion and is practically the same as the conclusion of the work. It should be removed from the discussion.

The part was removed from the discussion.

>Conclusion

The conclusion is good, but it must respond clearly to the proposed objective.

The conclusion was slightly rephrased in order to be more clearly related to the objectives.